# Evaluating and Revising the Digital Citizenship Scale

Randy Connolly [1] and Janet Miller [2],*

1   Department of Math and Computing, Mount Royal University, 4825 Mount Royal Gate SW,
    Calgary, AB T3E 6K6, Canada
2   Student Counselling, Mount Royal University, 4825 Mount Royal Gate SW, Calgary, AB T3E 6K6, Canada
*   Correspondence: jbmiller@mtroyal.ca

**Abstract:** Measuring citizen activities in online environments is an important enterprise in fields as diverse as political science, informatics, and education. Over the past decade, a variety of scholars have proposed survey instruments for measuring digital citizenship. This study investigates the psychometric properties of one such measure, the Digital Citizenship Scale (DCS). While previous investigations of the DCS drew participants exclusively from single educational environments (college students, teachers), this study is the first with a survey population ($n$ = 1820) that includes both students and the general public from multiple countries. Four research questions were addressed, two of which were focused on the validity of the DCS for this wider population. Our results suggest refining the 26-item five-factor DCS tool into an abbreviated 19-item four-factor instrument. The other two research questions investigated how gender, generation, and nationality affect DCS scores and the relationship between the different DCS factors. While gender was found to have a minimal effect on scores, nationality and age did have a medium effect on the online political activism factor. Technical skills by themselves appear to play little role in predicting online political engagement; the largest predictor of online political engagement was critical perspective and a willingness to use the Internet in active ways beyond simply consuming content.

**Keywords:** citizen participation; digital citizenship; measurement; online activism; digital citizenship

## 1. Introduction

Since its introduction almost three decades ago, the Internet has been the subject of near-constant scrutiny for its potential effects on our knowledge, social relations, and democratic institutions. One especially salient question is how these digital affordances are transforming the nature of citizenship, which has moved beyond the traditional dutiful norms of voting, party membership, and news awareness, to include a wide range of participation activities, such as civic volunteerism, protesting, and Internet-based political mobilization. These forms of online political involvement are collectively referred to here as digital citizenship, which has been an active area of research within educational, political, and communication communities.

It is becoming increasingly important to include a measure of digital citizenship when assessing how not only students but also citizens in general digitally engage with their social and political milieus, how political involvement among groups or countries compares, and how individuals interact in online environments to effect social and political change. This study investigates the psychometric properties of one such measure, the Digital Citizenship Scale (DCS), originally developed by Choi, Glassman, and Cristol [1]. A recent article published in Informatics included the DCS in its analysis of the three most common instruments for measuring digital citizenship [2]. They concluded that, "more replications, comparatives, and studies are needed in the use of digital citizenship assessment tools", particularly those that "expand the study sample as far as possible in the general population" [2] (p. 8).

While previous investigations of the DCS drew participants exclusively from educational environments (college students, teachers), this study is the first that uses the DCS with a survey population ($n = 1820$) that includes the general public from multiple countries. Our analysis examined the validity of the DCS with this wider population and investigated how gender, generation, and nationality affect DCS scores.

### 1.1. Contemporary Citizenship and Engagement

Within political science, citizenship is typically understood as a shared set of expectations about how members of a society engage in the political realm [3]. These expectations typically involve political participation, attainment of political knowledge, and respect for the rights of others. While some [4–6] worry about declines in formal political participation rates, in contrast, others argue that participation rates are being maintained but that the nature of political participation has transformed [7,8]. This scholarly interest in new forms of political participation also coincided with the wide-scale adoption of social networking platforms. Early on, arguments were often made that the political participation of the young had migrated to these digital channels [9,10]. However, even before the COVID-19 pandemic, it had become clear that political participation in all age groups had become at least partly digitally mediated [11–13].

### 1.2. Digital Citizenship and Its Measurement

This broadening of possible political participation repertoires has certainly made the measurement of contemporary citizenship more theoretically uncertain, as there is considerable overlap between the related research areas of digitally networked participation, digital citizenship, and digital literacy [14]. Even if one focuses just on digital citizenship, there are multiple distinct approaches to measurement. The most common of these is focused on "norms of appropriate responsible behavior with regards to technology use" [15]. It is oriented towards K-12 education with a focus on computer literacy, responsible online behaviour, and appropriate use of technology [16–19]. However, as Emejulu and McGregor [20] noted, this understanding of digital citizenship does little to help us "critically understand citizen's social relations with technology and the 'digital' and, in fact, obscures and silences the particular politics embedded within digital citizenship." Westheimer and Kahne [21] argued that education initiatives need to integrate three types of contemporary citizenship understandings: the personally responsible citizens, the participatory citizens, and the justice-oriented citizens (see also Choi and Cristol [22]). As Heath [23] observed, theorizing an educational version of digital citizenship that is stripped of political attributes leaves one with a citizenship concept that is ultimately a mere "platitude".

Other scholars, in contrast, have been more willing to encompass the political in their approach to measuring digital citizenship. One of these is the Digital Citizenship Scale (DCS) of Choi, Glassman, and Cristol [1], which provides a survey instrument for assessing the abilities, perceptions, and levels of political participation of individuals in their online activities. Their 26-item questionnaire has five distinct factors: Internet political activism (IPA, 9 items), technical skills (TS, 4 items), local/global awareness (LGA, 2 items), critical perspective (CP, 7 items), and networking agency (NA, 4 items); their study participants were from a mid-western American university, while a follow-up study by Choi, Cristol, and Gimbert [24] surveyed teachers in the United States. Others [2,25–27] followed the original study by using college students as their study participants.

### 1.3. The Current Study

This paper provides an additional instrument study replication but is the first to use it on both students and non-students across national boundaries. Our research questions, however, had a broader scope than just replication and comparison. Specifically, our questions were:

**RQ1**: Does the DCS retain its validity when used in different countries with roughly similar socio-economic development and in different generational cohorts?

**RQ2:** If the DCS is not valid for these expanded populations, can it be adjusted to create a better fit?

**RQ3:** To what extent do gender, nationality, and generation (age) affect the DCS?

**RQ4:** How do the other DCS factors relate to the IPA factor?

## 2. Materials and Methods

### 2.1. Participants

Two methods of recruitment were used. First, undergraduate students were invited to complete the survey, with participants being drawn from a midsize undergraduate university in Canada (*n* = 515), a large research university in Slovenia (*n* = 229), and a midsize university in Australia (*n* = 347). These institutions and countries were chosen for convenience. To broaden our participant pool beyond university students, additional participants were recruited through the SurveyMonkey Audience (SMA) platform (Canada, *n* = 302; Slovenia, *n* = 185; Australia, *n* = 242). Sampling participants from crowdsource populations (such as SMA) has become a well-respected research practice used in many fields [28,29].

In all, 1915 responses were gathered in late 2018, and 1820 responses with valid data from three countries were analysed (Canada, *n* = 817; Australia, *n* = 589; and Slovenia, *n* = 414). Demographic information about the samples is presented in Table 1. Participants self-selected one of seven pre-set age categories. Age categories were mapped to generation labels as follows: Gen-Z (18–20), Millennial/Gen-Y (21–25, 26–30), Gen-X (31–40, 41–50), and Boomer (51–60, 61+).

**Table 1.** Demographic Information.

| | Gender | | Age Groups (Generation Name) | | | | Total/% |
|---|---|---|---|---|---|---|---|
| | **Males** | **Females** | **Gen-Z** | **Millennial Gen-Y** | **Gen-X** | **Boomer** | |
| Canada | 437 | 380 | 343 | 195 | 143 | 136 | 817/44.9% |
| Australia | 213 | 376 | 93 | 143 | 218 | 135 | 589/32.4% |
| Slovenia | 178 | 236 | 241 | 45 | 95 | 33 | 414/22.7% |
| Total/% | 828/45.5% | 992/54.5% | 677/37.2% | 383/21.0% | 456/25.1% | 304/16.7% | 1820 |

### 2.2. Measures

This study made use of an online survey that included demographic questions and the original 26-item DCS. All items were answered using a 7-point Likert scale, ranging from 1 (strongly disagree) to 7 (strongly agree).

### 2.3. Statistical Analyses

The statistical analyses were conducted in four phases using the steps illustrated in Figure 1. First, to evaluate RQ1 (Research Question 1), we reproduced the explanatory factor analysis (EFA) on the full data set using the same parameters and approaches reported by Choi, Glassman, and Cristol [1]. The second phase of our analysis was focused on RQ2. Correlation results were reviewed, items were reassessed, and cross-national and cross-generational EFAs were run. From there, descriptive and comparative statistics were computed on the revised scale to assess RQ3, the effect of interactions between gender, nationality, and generation on the different scale scores. Lastly, in order to assess RQ4, regression and linear modeling were used to better understand the interaction effects of the different scale items.

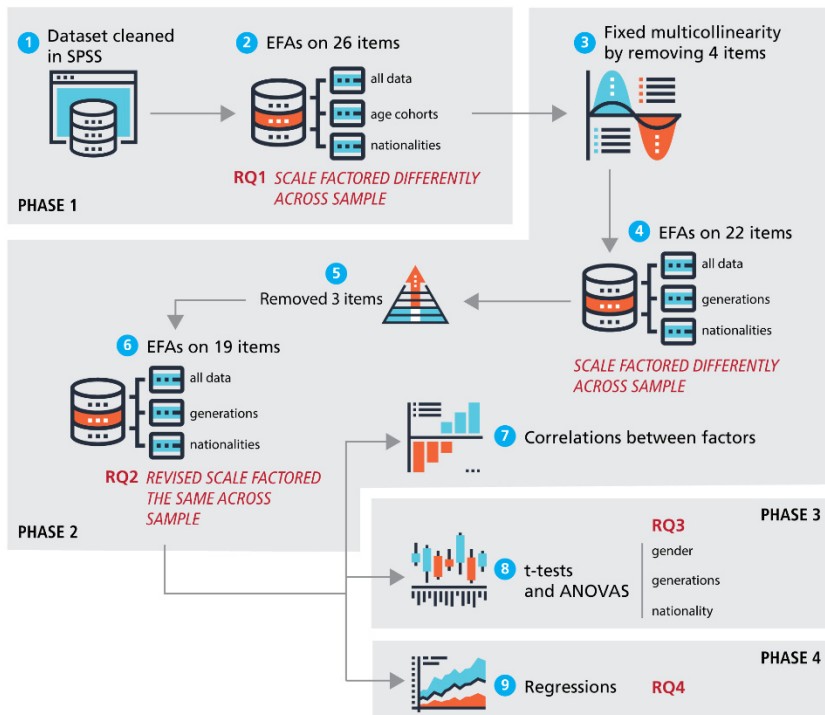

**Figure 1.** Analysis Process.

## 3. Results

### 3.1. Phase 1: Initial Exploratory Factor Analysis

This phase was focused on the question of whether the DCS is valid for different age cohorts and across nationalities. We began by reproducing the exploratory factor analysis reported in the original DCS. Our sample-to-variable ratio was 84:1, well above the recommended 30:1 ratio for factor analysis [30] and the 10:1 ratio reported in the original DCS study. A principal factor analysis was conducted on the original 26-item DCS with oblique rotation, which resulted in most of the same items grouped into the same five factors as the original DCS study. However, two IPA items (numbers 11 and 12 from the DCS) were found to fit best within the NA factor.

To investigate differences across nationality and age cohorts, the EFA results were compared across groups. For age, the data were split into two samples: those 30 or younger and those 31 and older. The 30 or younger sample's EFA results roughly replicated the original DCS EFA results, which gave us confidence to make broader comparisons with our data set. However, the EFA for the 31+ age sample varied substantially from the DCS EFA, with four questions (8, 9, 11, and 12) factored differently and with six items (8, 9, 12, 13, 20, and 25) exhibiting significant cross-listing.

To investigate the consistency of factors across nationalities, separate EFAs were calculated for the Canadian, Australian, and Slovenian responses. The Canadian data had two items (12 and 13) that factored differently in comparison to the original DCS, the Australian sample had seven items (6, 7, 8, 9, 12, 13, and 20), and the Slovenian data produced only four factors (instead of the five in the DCS) with four items (8, 9, 20, and 25) factored differently compared to the original. Cross-listing was also an issue for the Slovenian sample, with four items (11, 12, 20, and 25) exhibiting significant multiple factor loadings.

An additional problem was that for all of these EFAs, multicollinearity (as indicated by the determinant of the R-matrix value) was distorting the factor analysis: in our case, the value was 0.0000003. The common heuristic for this value [30] is that it should be larger than 0.00001. Our very small determinant value thus indicated that the original 26-item scale had too many items highly correlated with each other. That is, several of

the items in the original DCS were likely measuring the same phenomenon. In summary, then, the answer to RQ1 is that the original DCS was *not* valid for our multi-national, multi-generation data set.

### 3.2. Phase 2: Revised Exploratory Factor Analysis

Given the result of our first phase, the second phase was more exploratory and focused on whether the DCS could be adjusted so as to achieve validity across generations and nationalities. The process and analysis used in this phase can be found in the Appendix A and is also illustrated in Figure 1. In short, seven questions that were causing multicollinearity or that had low factor loading scores were removed. The resulting revised 19-item scale was then valid across generations and across the three countries in our sample. Table 2 (item numbers refer to the numbers from the original DCS) lists the results from our Revised Digital Citizenship Scale (DCS-R).

**Table 2.** Revised Digital Citizenship Scale (DCS-R) factor loadings.

| | Items | Factor 1 (IPA) | Factor 2 (TS) | Factor 3 (CP) | Factor 4 (NA) |
|---|---|---|---|---|---|
| *Factor 1: Internet Political Activism (IPA)* | | | | | |
| 20. | I organize petitions online about social, cultural, political, or economic issues online | 0.82 | | | |
| 18. | I work with others online to solve local, national, or global issues | 0.79 | | | |
| 17. | I work or volunteer for a political party via online methods | 0.78 | | | |
| 15. | I sometimes contact government officials via online methods | 0.75 | | | |
| 14. | I belong to online groups that are involved in political or social issues | 0.61 | | | |
| 19. | I sign petitions about social, cultural, political, or economic issues online | 0.47 | | | |
| *Factor 2: Technical Skills (TS)* | | | | | |
| 3. | I can use the Internet to find information I need | | 0.84 | | |
| 2. | I am able to use digital technologies to achieve the goals I pursue | | 0.79 | | |
| 4. | I can use the Internet to find and download applications that are useful to me. | | 0.77 | | |
| *Factor 3: Critical Perspectives (CP)* | | | | | |
| 23. | I think online participation is an effective way to make a change to something I believe to be unfair or unjust. | | | −0.83 | |
| 22. | I think online participation promotes offline engagement. | | | −0.74 | |
| 25. | I think I am given to rethink my beliefs regarding a particular issue/topic when using the Internet. | | | −0.66 | |
| 13. | I think online participation is an effective way to engage with political or social issues. | | | −0.56 | |
| 26. | I think the Internet reflects biases and dominance present in offline power structures. | | | −0.51 | |
| 21. | I am more socially or politically engaged when online than offline. | | | −0.51 | |
| 24. | I use the Internet in order to participate in social movement/change or protest. | | | −0.49 | |

**Table 2.** *Cont.*

| Items | | Factor 1 (IPA) | Factor 2 (TS) | Factor 3 (CP) | Factor 4 (NA) |
|---|---|---|---|---|---|
| *Factor 4: Networking Agency (NA)* | | | | | |
| 10. | Where possible, I comment on other people's writings in news websites, blogs, or SNSs I visit | | | | −0.88 |
| 9. | I post original messages, audio, pictures, or videos to express my feelings/thoughts/ideas on the Internet | | | | −0.70 |
| 11. | I regularly post thoughts related to political or social issues online. | | | | −0.51 |
| % Variance | | 38.2% | 14.0% | 7.9% | 5.3% |
| Eigenvalue | | 7.26 | 2.67 | 1.49 | 1.00 |
| Cronbach's alpha (all) | | 0.88 | 0.84 | 0.85 | 0.82 |
| Cronbach's alpha (Canada) | | 0.90 | 0.82 | 0.85 | 0.82 |
| Cronbach's alpha (Slovenia) | | 0.88 | 0.88 | 0.85 | 0.84 |
| Cronbach's alpha (Australia) | | 0.85 | 0.82 | 0.85 | 0.84 |

NOTE: The seven questions from the original DCS that were excluded from the Revised Scale were: 1. I can access the Internet through digital technologies (e.g., mobile/smart phones, Tablet PCs, Laptops, PCs) whenever I want; 5. I enjoy communicating with others online; 6. I enjoy collaborating with others online more than I do offline; 7. I am more informed with regard to political or social issues through using the Internet; 8. I am more aware of global issues through using the Internet; 12. I express my opinions online to challenge dominant perspectives or the status quo with regard to political or social issues; 16. I attend political meetings or public forums on local, town, or school affairs via online methods.

Correlations between the four factors are presented in Table 3. While all of the factors had statistically significant correlations, given the relatively large size of our sample, even very weak correlations were likely to be statistically significant [31]. The weak correlations of TS likely indicate that one's CP, NA, and IPA scores are only very partially explained by the original scale's technical skill questions. The relatively strong correlations (but not so highly correlated that multicollinearity would be an issue) between CP, NA, and IPA were subsequently analysed in Phases 3 and 4 below. In sum, the results of our revised EFA indicate that the answer to RQ2 is that the DCS can be adjusted (by removing seven items) to ensure validity for different generations and nationalities.

**Table 3.** Pearson correlations between factors.

| | IPA | TS | CP | NA |
|---|---|---|---|---|
| Internet Political Activism (IPA) | - | | | |
| Technical Skills (TS) | −0.073 ** | - | | |
| Critical Perspectives (CP) | 0.611 ** | 0.154 ** | - | |
| Networking Agency (NA) | 0.631 ** | 0.091 ** | 0.537 ** | - |

** Correlation is significant at the 0.01 level (2-tailed).

### 3.3. Phase 3: Descriptive and Comparative Statistics

Our third research question was concerned with the effect of gender, nationality, and generation on the DCS. The means (shown in Table 4) for the factors in our DCS-R exhibited similar central tendencies to those reported for the original DCS. Reported TS was very high across genders, nationalities, and generations, while IPA was similarly low across all groups. Independent *t*-tests (see Table 5) examined the role of gender in the differences in means in the four factors. Because our sample size was relatively large, the differences in IPA scores between genders were flagged as significant ($p < 0.05$), but as can be seen in Table 5, when Cohen's *d* was calculated, the effect size of the mean difference was very small.

**Table 4.** Descriptive statistics.

|  | IPA | TS | CP | NA |
|---|---|---|---|---|
| All | 2.76 (1.41) | 6.38 (0.80) | 4.15 (1.18) | 3.57 (1.58) |
| *Gender* |  |  |  |  |
| Male | 2.83 (1.51) | 6.37 (0.82) | 4.12 (1.19) | 3.56 (1.64) |
| Female | 2.69 (1.31) | 6.39 (0.78) | 4.17 (1.17) | 3.57 (1.53) |
| *Nationality* |  |  |  |  |
| Canada | 2.60 (1.39) | 6.42 (0.72) | 4.18 (1.14) | 3.58 (1.56) |
| Australia | 3.19 (1.41) | 6.41 (0.75) | 4.41 (1.17) | 3.84 (1.62) |
| Slovenia | 2.43 (1.30) | 6.28 (0.99) | 3.73 (1.17) | 3.14 (1.47) |
| *Generation* |  |  |  |  |
| Gen-Z (18–20) | 2.34 (1.36) | 6.48 (0.68) | 4.08 (1.19) | 3.27 (1.50) |
| Gen-Y/Millennial (21–30) | 2.85 (1.30) | 6.39 (0.83) | 4.26 (1.09) | 3.61 (1.49) |
| Gen-X (31–50) | 3.16 (1.43) | 6.37 (0.86) | 4.25 (1.18) | 3.92 (1.60) |
| Boomer (51+) | 2.98 (1.40) | 6.19 (0.80) | 4.04 (1.18) | 3.57 (1.71) |

Note: columns show mean (1–7) followed by standard deviation in parentheses.

**Table 5.** Independent *t*-tests on gender.

|  | t | df | p | 95% CI | Cohen's d | Effect Size |
|---|---|---|---|---|---|---|
| IPA | 2.07 | 1777 | 0.039 * | 0.007, 0.270 | 0.098 | Very Small |
| TS | −0.72 | 1805 | 0.470 | −0.102, 0.047 | −0.034 | Tiny |
| CP | −0.99 | 1784 | 0.321 | −0.166, 0.054 | −0.048 | Tiny |
| NA | −0.22 | 1795 | 0.822 | −0.164, 0.130 | −0.010 | Tiny |

* Significant at the 0.05 level.

What about the effect of nationality and generational cohort on the factor means? These differences in country and generation for each factor were first visualized (as shown in Figure 2), and then ANOVAs were used to determine whether the differences between the factor means were statistically significant. As shown in Table 6, nationality and generational cohort were statistically significant for all four factors.

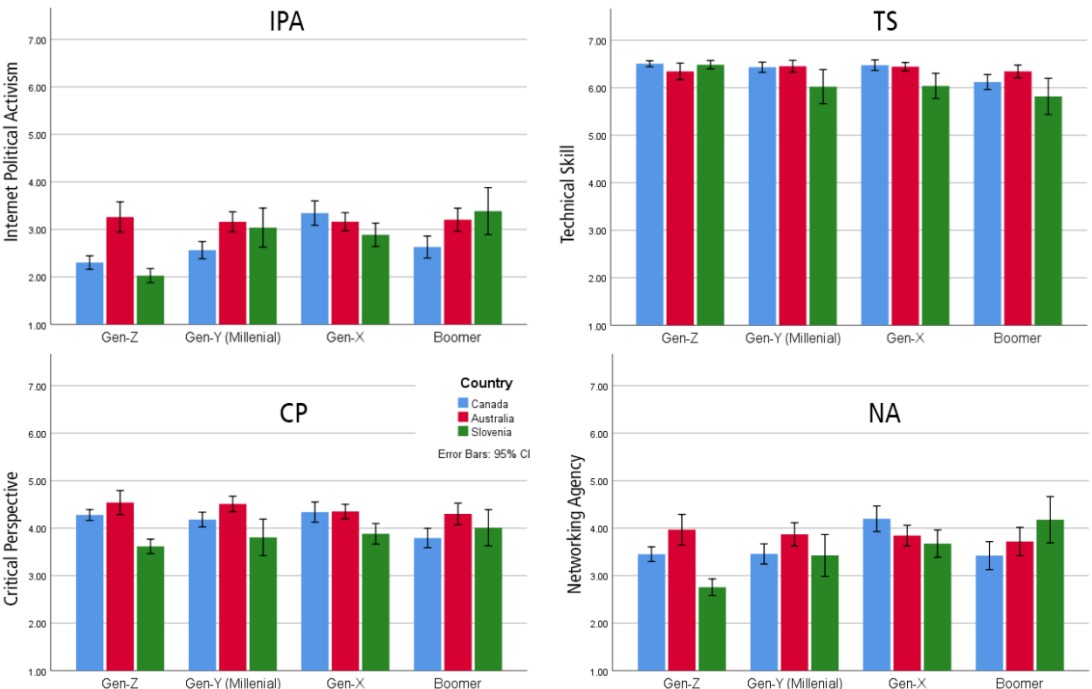

**Figure 2.** Generation and Nationality Across all Four Factors.

**Table 6.** Summary of ANOVA results of nationality and generation on DCS-R factors.

| | IPA (Internet Political Activism) | | | | TS (Technical Skill) | | | |
|---|---|---|---|---|---|---|---|---|
| Source | *df* | *F* | *p* | $\eta_P^2$ | *df* | *F* | *p* | $\eta_P^2$ |
| Nationality | (2, 1790) | 45.07 | 0.000 * | 0.048 ~ | (2, 1817) | 4.66 | 0.010 * | 0.005 |
| Generation | (3, 1790) | 69.81 | 0.000 * | 0.059 ~ | (3, 1817) | 9.15 | 0.000 * | 0.015 |
| | **CP (Critical Perspectives)** | | | | **NA (Networking Agency)** | | | |
| Source | *df* | *F* | *p* | $\eta_P^2$ | *df* | *F* | *p* | $\eta_P^2$ |
| Nationality | (2, 1797) | 42.39 | 0.000 * | 0.045 ~ | (2, 1808) | 24.23 | 0.000 * | 0.026 |
| Generation | (3, 1797) | 3.87 | 0.009 * | 0.006 | (3, 1808) | 15.95 | 0.000 * | 0.026 |

* Significant at the 0.05 level; ~ medium effect size [31].

The effect size of the differences, indicated by $\eta_P^2$ (partial eta-squared) in Table 6, was very small for most of the interactions. However, the effect size approached "medium" for generation and country on IPA as well as for country on CP. Thus, the answer to RQ3 was: gender did not appreciably affect one's DCS score, but generation and nationality did affect IPA and CP scores in a significant way.

*3.4. Phase 4: Regressions and Modelling*

Our fourth research question was focused on the relationship between the four different citizenship factors. In the original DCS study, the authors visualized their five factors as a triangle of ascending factors, with the lower factors as necessary building blocks for the higher factors. In that study, CP (critical perspectives) was at the pinnacle of the triangle, with IPA (Internet political activism), NA (network agency), LGA (local/global awareness), and TS (technical skills) grouped progressively under it. In their revised study [24], the positions of IPA and CP were reversed, showing that IPA is a more-difficult-to-achieve form of digital citizenship. Furthermore, they argued that because IPA requires TS, NA, and CP, but the reverse is not the case, this indicates that these three factors may merely be a means to an end. Indeed, if one understands digital citizenship as principally a political construct, we should not be surprised by the view that IPA is digital citizenship. Using data from our DSC-R, we tried to investigate the validity of this reasoning using hierarchical multiple regression analysis.

The full model of gender, country, generation, TS, NA, and CP to predict IPA (Model 4) was statistically significant ($R^2 = 0.560$, $F(6, 1718) = 363.74$, $p < 0.005$, adj. $R^2 = 0.558$). A series of regressions were run to determine the predictive ability of each of these on IPA. By themselves (Model 1), neither gender, nationality, nor TS was statistically significant. However, generation, NA, and CP were statistically significant in the prediction ($p < 0.05$). Regression coefficients can be found in Table 7.

**Table 7.** Hierarchical multiple regression analysis for IPA.

| | IPA | | | | | | | |
|---|---|---|---|---|---|---|---|---|
| | **Model 1** | | **Model 2** | | **Model 3** | | **Model 4** | |
| Variable | *B* | β | *B* | β | *B* | β | *B* | β |
| Constant | 21.488 | | 16.653 | | 10.566 | | 4.556 | |
| Gender | −0.834 | −0.050 | −0.805 | −0.048 * | −0.919 | −0.055 ** | −1.058 | −0.063 ** |
| Nationality | 0.305 | 0.046 | 0.306 | 0.046 | 0.498 | 0.074 ** | 0.657 | 0.098 ** |
| TS | −0.72 | −0.068 | −0.488 | −0.046 | −1.204 | −0.113 ** | −1.63 | −0.153 ** |
| Generation | | | 1.547 | 0.205 ** | 0.879 | 0.116 ** | 0.998 | 0.132 ** |
| NA | | | | | 3.335 | 0.627 ** | 2.143 | 0.403 ** |
| CP | | | | | | | 3.019 | 0.426 ** |
| $R^2$ | | 0.009 | | 0.051 | | 0.432 | | 0.560 |
| *F* | | 5.256 ** | | 22.908 ** | | 261.404 ** | | 363.744 |
| $\Delta R^2$ | | 0.009 | | 0.042 | | 0.381 | | 0.128 |
| $\Delta F$ | | 5.256 ** | | 75.183 ** | | 1153.963 ** | | 497.748 ** |

* $p < 0.05$ level, ** $p < 0.001$ level; B = unstandardized regression coefficient; β = standardized coefficient.

The measures of most importance for the interpretation of any hierarchical multiple regression are $R^2$ (which here represents the variation in IPA explained by the variables) and $\Delta R^2$ (which represents how much explanatory power has been added by including additional variables). For instance, in Model 1, the variables gender, nationality, and TS explained less than 1% of the variance in IPA; the addition of the generation variable in Model 2 added 4 more percent. Models 3 and 4 showed that NA and CP provided the most important explanation for IPA scores.

Thus, the answer to RQ4 is that NA and CP together are the most important predictor of being politically active online, while TS is relatively inconsequential. Indeed, our results indicate that those who are most politically active online are those with the highest critical perspective scores; similarly, those who are not politically active online are much more likely to have low critical perspective scores.

## 4. Discussion

### 4.1. Revised DCS Improvements

Choi, Glassman, and Cristol [1] reintroduced political questions into the measurement of digital citizenship. Their 26-item scale focused on political behaviours in the online world, moving away from the literacy and behavioural focus seen in earlier digital citizenship measures (e.g., [16]). Our study sought first to validate the original DCS five-factor model and test its reliability across samples drawn from various countries, age groups, and sources. The analysis conducted here revealed a 19-item revised DCS that retained its integrity across age groups, nationalities, and genders.

### 4.2. Is IPA Equivalent to Digital Citizenship?

Recent citizenship research has focused on the link between a wide range of participation activities and citizenship in general [11–13,21,32]. In this literature, citizenship is characterized precisely (and thus measurably) by the types of participation activities citizens engage in. That is, an active citizen is someone who is participating in one or more political participation activities. Or, more concisely, citizenship (digital or not) = political participation (broadly understood).

The taxonomy of Theocharis and Van Deth [32] is especially helpful here. They found that citizens engage (or do not engage) in six distinct forms of political participation activities: voting, digitally networked participation, institutionalized participation, protest, civic participation, and consumerist participation. The IPA (Internet political activism) scale in the DCS is clearly measuring the digitally networked participation identified by Theocharis and Van Deth [32]. The DCS and DCS-R are not measuring citizenship in general because that would have required also measuring these other participation modalities. Instead, we can say digital citizenship = just digitally networked participation (captured as IPA in the DCS).

Thus, through the IPA factor, the DCS-R does tell us something interesting about digital citizenship. As revealed in this study, technical skills, gender, and nationality appear to play a very minimal role in predicting a participant's IPA. Generation played a more statistically significant (albeit small) role in that older respondents were more likely than younger ones to use the Internet for political actions (though in a recent examination of digital vs. non-digital activism, [33] found the opposite interaction). Interestingly, our pre-COVID-19, late 2018 data showed that technical skills and networking agency were statistically equivalent between older and younger generational cohorts. While this might mean the relationship between age and technical skills is no longer salient, it might also simply reflect sampling bias (an online survey is going to attract those already comfortable with the Internet) or a lack of discernment in the TS questions in the DCS. Regardless, what was different in the older groups was not technical know-how or networking usage but an interest in politics and a willingness to use that interest for political/social engagement.

It is also important that these results varied across all three nationalities. Australians had significantly higher IPA scores than Canadians or Slovenians. Our Slovenian sample

had significantly lower IPA scores than did the Canadians. This indicates that researchers should be very hesitant about the generalizability of digital citizenship results pulled from a single country.

### 4.3. The Role of Critical Perspective

As revealed in Table 7 in the Results section, networking agency (NA) and critical perspectives (CP) are much more predictive of IPA than generation, gender, technical skills, or nationality, a finding that would be congruent with the expectations of Choi's [33] conceptual analysis of the digital citizen education literature. This finding about NA's importance is to be expected. Given that our IPA items are assessing a respondent's willingness to be politically active on the Internet, a necessary precondition would be a general willingness to be active (e.g., posting as opposed to simply viewing) in online environments, which is what NA measures. The regression results in this study support this assumption.

One of the most important findings in our regression analysis was the importance of critical perspective to active political uses of the Internet, regardless of nationality. High IPA scores were clearly related to high CP scores; similarly, low IPA scores were also related (but a little less clearly) to low CP scores. While we cannot import causal direction to this relationship, it is plausible that efforts to increase citizens' critical awareness may make them more likely to be politically engaged while online. Of course, the relationship might actually be the reverse: being politically active online is what increases one's critical perspective. Either way, these results indicate that further exploring the relationship between critical perspectives and political activity on the Internet may be a fruitful avenue for further quantitative study.

### 4.4. Limitations

While the DCS-R factored consistently across the three countries investigated, there was statistically significant variability in IPA and CP scores between these populations, and thus, one should have some hesitation about generalizing these results to other countries. As with any self-reporting questionnaire, results are influenced by bias and limited by the degree to which individuals are aware of their actual behaviours. Further, using an online survey to capture digital citizenship behaviours likely skews the results, especially for the NA and TS factor scores. Similarly, this sample was not balanced by socio-economic factors, which might also have skewed our results. Further, the Slovenian student sample used a translated version of the survey, while the adult sample used the English version, a difference that may further affect generalizability.

It would be ideal if each factor in the revised scale had the same (or close to the same) number of items. The uneven number of items for each factor (IPA: 6; TS: 3; CP: 7; NA: 3) is a vestige of the original DCS, which was also uneven in the number of items per factor.

Finally, these survey data were collected in late 2018, before the widespread normalization of online approaches to work and social life necessitated by the COVID-19 pandemic. However, just because general attitudes to digital infrastructure and politics may have changed during the pandemic, this does not necessarily mean the analytic validity of the DCS-R has been compromised.

## 5. Conclusions

Measuring citizen activities and attitudes continues to be an important enterprise in fields as diverse as political science, informatics, and education. This study is the first usage of the DCS across a wider population, one that included multiple nationalities and age cohorts. The results of this study showed that the original DCS was not valid across generations and nationalities. However, refining the original 26-item five-factor DCS tool into a 19-item four-factor instrument produced a strong consistency of factor loading across demographic populations and nationalities. Gender differences were minimal, but nationality and generation affected general political activism online. Technical skills by

themselves appear to play little role in predicting online political engagement; the largest predictor was critical perspective and a willingness to use the Internet in active ways. As a consequence of these findings, we believe the Revised Digital Citizenship Scale (DCS-R) may be an improvement of the original DCS, and that it can be used for cross-national, cross-generational comparisons.

**Author Contributions:** Conceptualization, R.C. and J.M.; methodology, R.C. and J.M.; formal analysis, R.C. and J.M.; investigation, R.C. and J.M.; resources, R.C. and J.M.; data analysis, R.C.; writing—original draft preparation, R.C. and J.M.; writing—review and editing, R.C. and J.M.; visualization, R.C.; project administration, R.C. and J.M.; funding acquisition, R.C. and J.M. All authors have read and agreed to the published version of the manuscript.

**Funding:** This research was funded by Mount Royal University through an internal research grant (Award #101276) and a Faculty of Science and Technology research grant.

**Institutional Review Board Statement:** This study was approved by the Human Ethics Research Board of Mount Royal University (HREB #101409).

**Informed Consent Statement:** Informed consent was obtained from all subjects involved in the study.

**Data Availability Statement:** The data presented in this study are openly available in Mendeley Data at https://data.mendeley.com/datasets/2v5fz56bjk/1, doi: 10.17632/2v5fz56bjk.1. Accessed on 17 August 2022.

**Conflicts of Interest:** The authors declare no conflict of interest.

## Appendix A. EFA Examination Procedure

Following a procedure recommended by Field [31], we constructed a correlation matrix and removed four items from the DCS whose correlation values were high with either one other item ($r > 0.8$), two other items ($r > 0.7$), or three other items ($r > 0.6$). Using this approach, we removed items 1, 8, 12, and 16. Doing so resulted in an acceptable multicollinearity value (determinant of 0.0000583).

EFAs were re-run on the remaining 22 items, resulting in a reduced four-factor model compared to the five factors in the original DCS. For comparison purposes, we also ran both principal component analysis and principal factor analysis with orthogonal rotation (varimax), both of which also resulted in four factors. One item (item 7) was removed based on the same criteria as the original DCS study (i.e., factor loading < 0.40), which meant that our scale then consisted of 21 items. However, the factor loadings continued to vary unacceptably between nationalities and generations.

Attention was drawn to items 5 and 6, which had the lowest factor loading scores (just above the 0.4 threshold). In the original DCS study, the authors noted that item 6 also had low loadings in their confirmatory factor analysis, but they decided to retain it because of "its theoretical importance to the scale as a whole." Arguably, items 5 and 6 are measuring something quite different from the other two items in the NA factor (items 9 and 10), which measure active or expressive usage of the Internet. Items 5 and 6 instead measure the enjoyment of or preference for online communication/collaboration. As an experiment, we tried following Jolliffe's criterion [34], which recommends retaining factors whose eigenvalue is above 0.70; in doing so, we ended up with five factors, with items 9 and 10 the sole scale items in the fourth factor and items 5 and 6 also on their own in the fifth factor, indicating that our intuitions about these four questions are likely correct. It is these authors' opinion that neither enjoying online communication nor preferring it to offline communication are necessarily expressions of networking agency. As such, these two items (5 and 6) were candidates for removal from the DCS. Eliminating these two items had a very favourable effect when we re-ran our EFAs with the revised 19-item scale. We then ended up with the same factoring across our different age groups and different countries.

The Kaiser–Meyer–Olkin measure verified the sampling adequacy for the analysis (KMO = 0.923). Bartlett's Test of Sphericity provided a significant value ($p = 0.00$). Anti-

image correlation values for individual items were all greater than 0.74 (indeed, all but three were above 0.9), which is well above the acceptable limit of 0.5. Four factors had eigenvalues over Kaiser's criterion of 1 and, in combination, explained 65.33% of the variance. The scree plot similarly showed four factors above the point of inflexion. Parallel analysis [35] using Monte Carlo PA also confirmed that four factors should be retained.

The proportion of non-redundant residuals above 0.05 using the revised 19-item scale was 3% (6 items). This value indicates the global difference between the correlation matrix produced by the actual data and that produced by the factor model. Ideally, this value is as close to zero as possible: given the size of our sample, the number of items, and the number of factors, this is an excellent result. In comparison, in our EFI using the original 26-item scale, the proportion of non-redundant residuals above 0.05 was 20% (67 items), which is acceptable but indicates that it did not model the actual data nearly as well as the revised 19-item scale. Item and factor reliability scores using Cronbach's alpha were also found to be high with the revised scale.

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
