# Peer review of "Evaluating and Revising the Digital Citizenship Scale"

_informatics, doi:10.3390/informatics9030061_

Round 1

Reviewer 1 Report

Review of “Evaluating and Revising the Digital Citizenship Scale”

1. The paper investigates the psychometric properties of the Digital Citizenship Scale (DCS) through a large population survey.

2. The Abstract and Keywords are correct. I suggest the inclusion of ‘Digital Citizenship Scale’ in the keywords.

3. The paper is very well-written and formally correct.

4. Introduction is excellent with relevant literature and RQ.

5. Materials and methods are ok. I only have some doubts concerning the use of SurveyMonkey Audience for recruitment, but it was already tested by other authors. Figure 1 is perfect.

6. Results are well-detailed and clear.

7. Discussion is very interesting and presents an acknowledgement of the recruitment problem: ‘sampling bias (an online survey is going to attract those already comfortable with the internet)’. Limitations also address this problem.

8. Conclusions are ok, but they could have included a straight answer to each of the RQ.

9. References are ok.

Congratulations for the excellent work!

Author Response

Please see letter attached.

Reviewer 2 Report

The last article by Moonsun Choi contrasts three approaches to the concept of citizenship that could enrich the introduction of the manuscript.

Moonsun Choi & Dean Cristol (2021) Digital citizenship with intersectionality lens: Towards participatory democracy driven digital citizenship education, Theory Into Practice, 60:4, 361-370, DOI: 10.1080/00405841.2021.1987094

No envidences or data are shown showing why 7 items have been removed. For example item 1 does not factor differently by age or nationality groups, why is it removed, where are the results of the high correlation of the seven items?

To show the coherence of the reduction of items and factors, and given that in the case of Slovenia a translation of the questionnaire was made for part of the sample, it would be convenient to calculate the reliability of each factor by country as well (include in Table 2).

The results of the table with the correlation matrix should also be clarified, namely the correlation between IPA and TS is negative. Also all the results are significant ** although there are important differences according to the result of each statistical coefficient, is everything correct?

In this article the number of items of the same Choi's questionnaire is reduced... could we contrast or compare?

Yoon, S.; Kim, S.; Jung, Y. Needs Analysis of Digital Citizenship Education for University Students in South Korea: Using Importance-Performance Analysis. Educ. Technol. Int. 2019, 20, 1-24. [Google Scholar].

In this article Choi discusses Four Categories of the Concept of Digital Citizenship. Could you comment on the relationship with the four factors found.

Choi, M. (2016). A Concept Analysis of Digital Citizenship for Democratic Citizenship Education in the Internet Age. Theory & Research in Social Education, 44(4), 565-607.

For clarity, a regression analysis (stepwise mode) is recommended.

Given the relevance of IPA or online activism it is recommended to consult this short paper with insternational results of a brief scale of online activism for the final discussion.

J. S. Fernández-Prados, A. Lozano-Díaz, A. Ainz-Galende and R. Rodríguez-Puertas, "Intergenerational Digital and Democratic Divide: Comparative Analysis of Unconventional and Digital Activism around the World," 2021 9th International Conference on Information and Education Technology (ICIET), 2021, pp. 466-470, doi: 10.1109/ICIET51873.2021.9419635.

Errors:

Page 2 line 47 "counties" should be "countries".

Unify Probability notes:

*p < .05; **p < .01.

(no zeros before the decimal point)

Author Response

Please see the attached letter for details. Thank you.
